# Targeting Lung–Gut Axis for Regulating Pollution Particle–Mediated Inflammation and Metabolic Disorders

**DOI:** 10.3390/cells12060901

**Published:** 2023-03-15

**Authors:** Tzu-Yu Cheng, Chih-Cheng Chang, Ching-Shan Luo, Kuan-Yuan Chen, Yun-Kai Yeh, Jing-Quan Zheng, Sheng-Ming Wu

**Affiliations:** 1Division of Cardiovascular Surgery, Department of Surgery, Wan Fang Hospital, Taipei Medical University, Taipei 11696, Taiwan; tzuyucheng@tmu.edu.tw; 2Division of Cardiology, Department of Internal Medicine, School of Medicine, College of Medicine, Taipei Medical University, Taipei 11031, Taiwan; 3Division of Pulmonary Medicine, Department of Internal Medicine, Shuang Ho Hospital, Taipei Medical University, New Taipei City 23561, Taiwan; 09005@s.tmu.edu.tw (C.-C.C.); 16217@s.tmu.edu.tw (C.-S.L.); 14388@s.tmu.edu.tw (K.-Y.C.); 19354@s.tmu.edu.tw (Y.-K.Y.); 16044@s.tmu.edu.tw (J.-Q.Z.); 4Division of Pulmonary Medicine, Department of Internal Medicine, School of Medicine, College of Medicine, Taipei Medical University, Taipei 11031, Taiwan; 5TMU Research Center for Thoracic Medicine, Taipei Medical University, Taipei 11031, Taiwan; 6International Ph.D. Program in Cell Therapy and Regenerative Medicine, College of Medicine, Taipei Medical University, Taipei 11031, Taiwan; 7Graduate Institute of Clinical Medicine, College of Medicine, Taipei Medical University, Taipei 11031, Taiwan

**Keywords:** cigarette smoking, lung–gut axis, inflammation, metabolic disorder, particulate matter

## Abstract

Cigarette smoking (CS) or ambient particulate matter (PM) exposure is a risk factor for metabolic disorders, such as insulin resistance (IR), increased plasma triglycerides, hyperglycemia, and diabetes mellitus (DM); it can also cause gut microbiota dysbiosis. In smokers with metabolic disorders, CS cessation decreases the risks of serious pulmonary events, inflammation, and metabolic disorder. This review included recent studies examining the mechanisms underlying the effects of CS and PM on gut microbiota dysbiosis and metabolic disorder development; one of the potential mechanisms is the disruption of the lung–gut axis, leading to gut microbiota dysbiosis, intestinal dysfunction, systemic inflammation, and metabolic disease. Short-chain fatty acids (SCFAs) are the primary metabolites of gut bacteria, which are derived from the fermentation of dietary fibers. They activate G-protein-coupled receptor (GPCR) signaling, suppress histone deacetylase (HDAC) activity, and inhibit inflammation, facilitating the maintenance of gut health and biofunction. The aforementioned gut microbiota dysbiosis reduces SCFA levels. Treatment targeting SCFA/GPCR signaling may alleviate air pollution–associated inflammation and metabolic disorders, which involve lung–gut axis disruption.

## 1. The Lung–Gut Axis in Lung and Metabolic Disease

The homeostasis of the gut microbiota and its metabolites act as a critical role in the alternation of the local and systemic immune system [1]. Accumulating studies point to an important interaction between the gut microbiota and the lung, termed the “lung–gut axis” [1]. Furthermore, increasing evidence declares its crucial role in maintaining host metabolic homeostasis and in the pathogenesis of lung diseases [2]. Particulate air pollution components, such as hazardous substances and metals that remain in the lungs, can lead to dysbiosis of the lung and gut microbiota, resulting in decreased lung function [3]. The role of the lung–gut axis is related to the action of microbiota and gut microbe-derived components and metabolites, such as lipopolysaccharide (LPS), metabolite trimethylamine (TMA) N-oxide (TMAO), and short-chain fatty acid (SCFA), as main regulators for the immune and metabolic systems [4,5,6]. SCFAs regulate immune homeostasis and systemic immune responses, such as modulation of pro-inflammatory or anti-inflammatory [7]. Notably, gut microbiota dysbiosis is one of the risk factors promoting insulin resistance (IR) in type 2 diabetes mellitus (T2DM) [8]. Current evidence suggests that the chemical constituents of the air pollutant mixture may affect the pathogenesis of T2DM [9]. However, there is a need to elucidate the detailed mechanisms by which air pollutants cause metabolic dysfunction, IR, and T2DM, including increased inflammation, oxidative stress, and endoplasmic reticulum stress [10]. Of note, emerging experimental and epidemiological findings highlight a key factor that exposure to pollutant particles induces alterations in the gut microbiota and systemic metabolism, which may contribute to glucose metabolism disorder, IR, and T2DM [11]. In this while, the regulation of gut microbiota homeostasis or probiotics administration may act as a vital role on the regulation of lung–gut axis. Indoor and outdoor air pollution such as cigarette smoking (CS) and ambient particulate matter (PM) increase lung dysfunction and risk of chronic obstructive pulmonary disease (COPD) and respiratory death [12]. This review includes recent studies on the effects of CS and ambient PM on gut microbiota dysbiosis, systemic inflammation, and metabolic diseases possibly due to disorder of the lung–gut axis. Furthermore, we discuss the potential prevention of particulate pollution-mediated dysregulation of the lung–gut axis and consequent microbial-derived metabolites that may reduce the risk of metabolic dysfunction and associated disease progression.

## 2. CS Exposure Induces Lung Inflammation and Diabetes Mellitus Progression

CS is a complex mixture including various compounds with suspended PM and gases [13]. Indoor PM_2.5_ (PM with aerodynamic diameter ≤ 2.5 μm) was the most reliable indicator in CS [14]. CS exposure can cause chronic inflammatory lung disease and lung infection and predispose individuals to acute lung injury [15]. Airway epithelial cells and alveolar macrophages exposed to CS alter inflammatory signaling leading to the recruitment of eosinophils, neutrophils, lymphocytes, and mast cells to the lung—depending on the activation of different signaling pathways, such as Nuclear factor-κB (NF-κB), c-Jun N-terminal kinase, p38, and STAT3 [16]. In addition, CS is also considered a critical risk factor for metabolic disorder and metabolic syndrome (MetS). Metabolic disorders occur when aberrant chemical reactions in the body alter normal metabolic processes. However, MetS encompasses a spectrum of metabolic abnormalities, including hypertension, obesity, IR, and atherogenic dyslipidemia, and is strongly associated with an increased risk of diabetic cardiovascular disease [17]. Notably, CS may induce MetS, which is characterized by reduced insulin sensitivity, increased IR and plasma triglyceride levels, and mediated hyperglycemia [18]. Several clinical studies have indicated a close link between MetS and lung disease [19]. MetS comprises a cluster of risk factors specific for lung inflammation diseases, such as obesity, high blood pressure, diabetes mellitus (DM), and chronic kidney disease (CKD) [20,21]. CS exposure is strongly associated with an increased risk of progression of various diseases, including DM, cardiovascular disease, and COPD. However, prolonged CS cessation can reduce these risks.

A meta-analysis of prospective observational studies study revealed that there is a linear dose-response relationship between CS and the risk of T2DM; however, the risk steadily declined after smoking cessation [22]. There was a dose-response relationship between HbA1c (blood sugar/glucose test) levels and CS exposure of current smokers, and there was an inverse association between the number of years since smoking cessation with HbA1c levels has been detected among ex-smokers [23]. The adverse effects of CS and hyperglycemia aggravate vascular damage in patients with DM [24]. Eliasson et al. reported a dose-dependent correlation between the per-day CS quantity and IR degree [25]. Acute CS can impair glucose tolerance and insulin sensitivity as well as increase serum triglyceride and cholesterol levels, blood pressure, and heart rate [26], whereas CS cessation can improve insulin sensitivity [27]. Moreover, CS is an independent predictor of T2DM, and CS cessation can reduce the risk of MetS [28]. Serious pulmonary events (SPEs) reduced exercise capacity and lung dysfunction and might be a clinical indicator of pre-DM or undiagnosed DM [24]. CS cessation can reduce SPE risk, morbidity, and mortality in people with DM. For instance, in DM mice, CS exposure was reported to accelerate edema and inflammatory progression, whereas CS cessation alleviated CS-mediated metabolic disorder and IR [27]. Also, DM is strongly correlated with pulmonary complications, including COPD, fibrosis, pneumonia, and lung cancer [29]. In patients with COPD combined with DM, metformin—the first-line drug for T2DM—can increase inspiratory muscle strength, leading to dyspnea and COPD alleviation as well as health status and lung function improvements [30,31]. Although the strong association of DM with inflammation-related chronic lung disease, particularly asthma and COPD, has been demonstrated epidemiologically and clinically, the underlying mechanism and pathophysiology remain unclear [32]. Numerous mechanisms have been reported; most of them have implicated the association of lung disease with DM-related inflammatory properties or pulmonary microvascular and macrovascular complications [33]. DM, combined with progression to pulmonary disease, is characterized as a chronic and progressive disease with high mortality and extremely few therapeutic options, possibly including metformin and thiazolidinediones [32,34,35]. Collectively, the complex consequences mediated by CS exposure lead not only to lung inflammation but also to the progression of MetS. In the next section, we continue to review emerging studies on CS-mediated systemic inflammation and metabolic diseases associated with gut microbiota dysbiosis.

## 3. Association of CS Exposure with Gut-Derived Microbiota and Inflammatory Bowel Disease

CS-related microbiota dysbiosis may be associated with numerous inflammatory lung diseases (including COPD, asthma, cystic fibrosis, and allergy) and metabolic disorders (such as IR, glucose metabolic disorder, hypertriglyceridemia, and DM) [14,36,37]. However, the relationship of CS with gut-derived microbiota biofunction and their dietary nutrient–derived metabolites warrants elucidation. Even with an improved understanding of the mechanisms underlying the progression of inflammation-associated lung disease, CS-associated metabolic disorder remains the leading cause of morbidity and mortality globally. Several epidemiological studies have indicated a high correlation between intestinal microbiota and the lungs—termed the lung–gut axis [1]. Gut microbiota dysbiosis is associated with inflammatory bowel disease (IBD), and it influences the gut epithelial barrier function and leads to increased immune response and chronic inflammatory disease and metabolic disorder pathogenesis. Moreover, IBD is strongly linked with COPD, DM, and gut microbiota dysbiosis [38,39]. Alterations in gut microbiota due to an imbalanced diet may lead to enhanced local and systemic immune responses. Gut microbiota dysbiosis has been linked to not only the loss of gastrointestinal tract function but also that of airway function, such as that in asthma and COPD [1]. Moreover, CS-elicited dysbiosis has a protumorigenic role in colorectal cancer. CS-associated gut microbiota dysbiosis alters gut metabolites and diminishes gut barrier function, activating oncogenic MAPK/ERK signaling in the colonic epithelium [40]. Taken together, these results indicate that CS mediates not only the dysbiosis of gut microbiota and the dysregulation of their metabolites but also systemic inflammation and metabolic dysregulation in the lung–gut axis. In addition, ambient PM-mediated systemic inflammation and metabolic disease may also be associated with dysregulation of the lung–gut axis, as reviewed later.

## 4. PM Exposure Triggers Lung Inflammation, Gut Microbiota Dysbiosis, and DM Progression

Chronic exposure to particulate pollution such as PM_2.5_ can also lead to decreased lung function, emphysematous lesions, and airway inflammation [41,42], and it accelerates CS-induced alterations in COPD progression [36]. Ambient air pollution is associated with decreased lung function and increased COPD prevalence in a large cohort study [12]. Furthermore, urban PM exposure markedly increased airway inflammatory responses through the activation of reactive oxygen species (ROS)-MAPK-NF-κB signaling [43]. Thus, PM exposure contributes to respiratory disease by triggering lung inflammation and increasing oxidative stress.

Notably, recent studies have linked PM exposure to intestinal disorders such as appendicitis, irritable bowel syndrome, and IBD [44,45]. In addition, the secretion of proinflammatory cytokines and intestinal permeability is increased in the small intestine of mice exposed to PM_10_ [46]. PM-mediated dysbiosis of the microbiota is also correlated with PM-mediated gut and systemic effects. Long-term exposure to PM, such as O_3_, NO_2_, SO_2_, PM_10_, and PM_2.5_, as well as traffic-related air pollution, has been shown to alter microbiota diversity [47]. Moreover, both PM_2.5_ and PM_1_ exposure are positively associated with the risks of fasting glucose impairment or T2DM and negatively associated with alpha diversity indices of the gut bacteria [48]. Through systematic database analysis, air pollution exposure is a leading cause of insulin resistance and T2DM. Besides, the association between air pollution and diabetes is stronger for particulate matter, nitrogen dioxide, and traffic-associated pollutants [10]. In a meta-analysis study, exposure to PM_2.5_ but not PM_10_ or NO_2_ is correlated with increased disease incidence in T2DM patients [49]. Particularly, chronic ambient PM_2.5_ exposure is associated with increased T2DM risk in several Asian populations exposed to high levels of air pollution [50]. Through global untargeted metabolomic analysis, several significant blood metabolites and metabolic pathways were identified associated with chronic exposure to PM_2.5_, NO_2_, and temperature; these included glycerophospholipid, glutathione, and sphingolipid propanoate as well as purine metabolism and unsaturated fatty acid biosynthesis [51]. In mice, PM_2.5_ exposure was reported to lead to increased oxidative stress, glucose intolerance, IR, and gut dysbiosis and impaired hepatic glycogenesis [52]. In rats with T2DM, short-term PM_2.5_ exposure significantly increased IR as well as the lung levels of inflammatory factors, such as interleukin (IL) 6, monocyte chemoattractant protein 1, and tumor necrosis factor (TNF) α [39]. High levels and prolonged periods exposure to concentrated ambient PM_2.5_-mediated gut dysbiosis was associated with the metabolic disorder and intestinal inflammation [53]. Taken together, these findings indicate that air pollution particles not only mediate the pathogenesis of lung inflammation disease but also increase gut microbiota dyshomeostasis and metabolic disorders, such as IR and DM.

## 5. CS and PM Exposure Mediates Systemic Inflammation and Metabolic Disorders Associated with Lung–Gut Axis Disruption

Emerging studies on the effects of particulate pollution, such as CS and PM exposure, on the systemic inflammation and metabolic disorder associated with the lung–gut axis. Exposure to CS and PM_2.5_ is a critical risk factor for intestinal dysfunction, which comprises intestinal microbiota dysbiosis, enhanced permeability of the mucosal barrier, and induce mucosal immune responses [54,55]. The toxic constituents of cigarette smoke include carboxylic acids, phenols, humectants, nicotine, polycyclic aromatic hydrocarbons, acetaldehyde, 1,3-butadiene, N-nitrosamines, benzene, aromatic amines, acrolein, and polyaromatics, all of which are inhaled into the lungs [56], followed by their absorption into the blood system and then into the gastrointestinal tract; this causes dysbiosis of gut microbiota through the inhibition of their bioactivity and thus alterations in the intestinal microenvironment [57]. Moreover, these toxic compounds induce lung oxidative stress, gut microbiota dysbiosis, intestinal dysfunction, and extreme systemic inflammation. CS-exposed mice were noted to have impaired gut barrier function and elevated serum LPS levels compared with those not exposed to CS [40]. After 24 weeks of PM (biomass fuel or motor vehicle exhaust) exposure, rats were noted to display lung inflammation, which progressed to COPD. Their gut microbiota demonstrated decreased microbial richness and diversity and decreased SCFA levels, but increased serum LPS levels were found [58]. In young patients with hypertension, the metal constituents of PM_2.5_ were noted to elevate blood pressure and increase the plasma levels of the microbial metabolite trimethylamine TMAO [59]. PM mediates gut barrier function loss, thereby increasing gut permeability and consequently resulting in the entry of bacteria, bacterial endotoxin, bacterial metabolite, or all leak into circulation [60]. As well, dysregulated gut epithelial barrier, gut microbiota dysbiosis, and accelerated gut microbial product translocation promote lung inflammation [61]. In fact, the harmful components carried by ambient PM may vary according to the level and duration of environmental pollution exposure, leading to different degrees of lung deposition, inflammation and injury. However, either CS or PM may worsen IBD, gut microbiota dysbiosis, and microbiota-derived metabolite alterations, thus mediating systemic inflammation and metabolic disorder development by increasing LPS and TMAO levels and reducing SCFA levels. Below, we highlight recent studies that provide insight into the implication of particulate pollution exposure in aggravating systemic inflammation and metabolic disease through complex interplaying processes, including lung inflammation and injury, gut microbiota dysbiosis, and the production of corresponding metabolites.

## 6. Mechanisms Underlying CS and PM Exposure Aggravate Inflammation and Development of Metabolic Disorder through the Lung–Gut Axis

CS or PM inhalation studies have mostly focused on the effects of air pollution on inducing lung inflammatory response, oxidative stress, and endothelial dysfunction; however, the effects of PM on gut microbiota and its role in metabolic disease pathogenesis are largely unknown. CS-mediated chronic airway inflammation is a major COPD pathogenesis driver [62,63]. CS also mediates gut microbiota dysbiosis [40]. Gut microbiota dysbiosis has a potential role in CS-related pathogenesis, such as that in metabolic disorders and DM. When in homeostasis, the gut microbiome has a high diversity of microbiota, such as bacteria from the phyla *Firmicutes*, *Bacteroidetes*, *Actinobacteria*, *Verrucomicrobia*, and *Proteobacteria* [64]. Smokers’ gut microbiome differs from that of nonsmokers. The gut microbiota of the majority of current smokers includes a lower number of bacteria from *Firmicutes* but a higher number of bacteria from *Bacteroidetes* [65]. After CS cessation, these individuals demonstrate an increase in *Actinobacteria* and *Firmicutes* populations but a decline in *Proteobacteria* and *Bacteroidetes* populations [66,67]. Specifically, observational and interventional studies have revealed that current smokers have increased *Proteobacteria* and *Bacteroidetes* populations but decreased populations of *Actinobacteria* and *Firmicutes* as well as *Bifidobacterium* spp. and *Lactococcus* spp.—along with low gut microbiome diversity [66,68]. Current smokers have also been reported to have increased numbers of *Proteobacteria* as well as of *Bacteroides* spp. and *Prevotella* spp. in their feces; however, this abundance has been observed to decrease after CS cessation, with an increase in the numbers of *Firmicutes* (*Clostridium coccoides*, *Clostridium leptum subgroup*, and *Eubacterium rectale*) and *Actinobacteria* (*Bifidobacterium*) [69,70]. CS or bidi smoking is correlated with *Erysipelotrichi* and *Catenibacterium* abundance in a dose-response manner, where CS exposure leads to the enrichment of the relative abundance of *Erysipelotrichi*–*Catenibacterium* [71]. Adult mice exposed to CS have been reported to exhibit significantly reduced *Firmicutes* abundance but significantly increased *Bacteroidetes* abundance. CS-exposed mice also demonstrated significant alterations; *Eggerthella lenta* population enrichment and *Lactobacillus* spp. population decline [40]. *Firmicutes* (gram-positive) represent one of the most abundant and exclusive phyla in the intestines [72]. CS patients with active Crohn’s disease may have significantly higher *Bacteroides*–*Prevotella* levels than do nonsmokers [69]. Moreover, increases in the population of bacteria causing proinflammatory pathway activation might cause CS-mediated deleterious effects [73]. As mentioned, CS may reduce *Firmicutes* and increase *Bacteroidetes* populations; nevertheless, CS cessation can enable the return of the populations to their original levels. CS reduces the number of the SCFA-producing bacterium *Porphyromonas gingivalis*, causing a disparity in a cigarette smoker’s SCFA profile [74]. CS reduces the levels of the dietary fiber–friendly SCFA-generating *Firmicutes* bacteria but not the levels of *Bacteroidetes* bacteria. A study suggested a novel treatment for emphysema: high-fiber diet administration followed by feces microbiota transplantation [75].

Notably, disruption of the microbiota resulted in lower SCFA levels after exposure to air pollution; this effect is linked with metabolic disorders. When rats were exposed to CS for 4 weeks, the cecal levels of SCFAs, such as acetic acid, propionic acid, butyric acid, and valeric acid, decreased significantly. Moreover, *Bifidobacterium* spp. significantly decreased, whereas pH in caecal contents significantly increased [76]. Intragastric administration of cigarette smoke condensate (CSC) in mice caused inflammation in the intestinal mucosa, which induced alterations in Paneth cell granules and reduced their bactericidal capacity [57]. CSC exposure caused an imbalance in the gut bacterial population, promoting bacterial infection and causing ileal damage in mice. Moreover, CSC significantly increased the abundance of bacteria from *Erysipelotrichaceae* but considerably reduced that of *Rikenellaceae*. A significant decrease was also noted in the abundance of *Eisenbergiella* spp. (from the family *Lachnospiraceae*), known for its butyrate generation capacity [77]. CS cessation-related alterations in the gut microbial ecosystem have, however, been linked to weight gain in mice [78]. The concentrations of SCFAs (i.e., acetate, butyrate, and propionate) were the lowest in mice with CS-induced emphysema; on the contrary, local SCFA levels were significantly higher in the emphysema with high-fiber (pectin and cellulose) diet group than in the untreated emphysema group [75]. In diesel exhaust particles-treated mice, dysbiosis of the gut microbiota population was associated with a dose-dependent decrease of SCFAs (butyrate and propionate) in cecal content [79]. Collectively, accumulating findings suggest that CS- or PM-mediated lung injury is closely associated with altered gut microbiota and low SCFA production.

The potential mechanisms underlying the effects of CS-mediated metabolic disorders include inducing gut microbiota dysbiosis, increased oxidative stress, disrupted intestinal tight junctions, and increased systemic inflammation. Notably, some CS-mediated gut microbiota changes are similar to those demonstrated during the progression of conditions such as IBD, IR, glucose metabolic disorders, and DM. Besides, metabolomic analysis has revealed that CS exposure increases the levels of bile acid metabolites, particularly taurodeoxycholic acid (TDCA), in the colons of mice. These mice also had an upregulated TDCA–mitogen-activated protein kinase-extracellular signal-regulated protein kinase 1/2 axis and damaged gut barrier function [40]. Intratracheally instillated diesel PM_2.5_ in mice led to significant increases in the numbers of bacteria from *Firmicute*, specifically those from the family *Enterobacteriaceae*, and decreases in those from *Bacteroidetes*, specifically bacteria from the family *Porphyromonadaceae* [80]. Long-term exposure to PM_2.5_ for 12 months in mice resulted in IR and impaired glucose tolerance, which was linked to the alternations of gut bacterial communities richness and gut microbiota composition [55]. PM_2.5_, therefore, mediates changes in the microbiome composition; this might lead to metabolic disorder pathogeneses. Further exploration of the mechanisms underlying CS- and PM-associated microbiota dysbiosis is warranted. These findings may aid in elucidating whether air pollution-related gut microbiota alterations contribute to CS- and PM-related inflammation and metabolic disorders. The potential effects of CS or PM exposure on the lung–gut axis are schematically represented in Figure 1. CS or PM inhalation leads to gut microbiota dysbiosis and consequently increases the levels of TMAO and LPS, leading to system inflammation and exacerbated metabolic disorders. Notably, SCFA/GPCR signaling may retain gut barrier function and suppress inflammation. In addition, dietary fiber fermentation by certain specific gut microbiota can increase SCFA production. Thus, reducing microbiota dysbiosis and its attendant products attenuates lung inflammation and cellular injury by the lung–gut axis, which may prevent systemic inflammation and metabolic dysregulation.

## 7. Gut Microbiota Dysbiosis in DM

Undoubtedly, disruption of the gut microbiota homeostasis is closely associated with the pathogenesis of DM. Through metagenomics and metaproteomics analysis of fecal samples, a comparative study revealed that the population of butyrate-producing *Faecalibacterium prausnitzii* was lower in individuals with pre-DM than in both normal glucose-tolerant individuals and those with treatment-naïve T2DM [81]. A pilot study revealed significant alterations in the gut microbial communities between T2DM patients and non-T2DM controls [82]; the results indicate that an increased *Bacteroides* population is independently associated with elevated LPS levels and decreased insulin sensitivity, which might be connected to the gut barrier function. According to clinical evidence, the populations of butyrate-generated bacteria from the phylum *Firmicutes*, specifically the families *Lachnospiraceae* and *Ruminococcaceae*, are significantly decreased in patients with T2DM [83,84]. Probiotic intervention has been noted to increase gut butyrate metabolism, reducing the levels of the intermediates of bilirubin and fatty acid oxidation, indicating metabolic improvement in patients with T2DM [85]. Taken together, these results provide a new direction to understand the interaction between gut microbiota dysbiosis and T2DM development and progression. Here, we focus on the potential impact of particulate pollution exposure on the lung–gut axis, which is also associated with DM progression. In addition, controlling dysregulation of the lung–gut axis may benefit from disease progression through several strategies described below.

## 8. Therapeutic Strategies for CS-Mediated Inflammation and Metabolic Disorders

The following approaches may facilitate the development of novel therapeutics targeting the lung–gut axis to ameliorate systemic inflammation, thus inhibiting CS-associated DM progression.

### 8.1. CS Cessation

The intensity of the negative impact of CS is strongly correlated with the daily CS amount and overall CS duration; CS cessation can nevertheless aid in reducing the risks of COPD, DM, and lung cancer. CS contains compounds that increase inflammation, reduce insulin sensitivity, and promote cell oxidative stress, all increasing the risk of DM. In patients with DM, CS cessation is mostly associated with enhanced glycemic control [86]. In Japanese patients with T2DM, hemoglobin A1c (HbA1c), as an indicator of DM, was noted to increase progressively with the number of cigarettes smoked per day and CS pack-years compared with never-smokers; the HbA1c decreased linearly with the years after CS cessation [87]. In another cohort study, 241 patients with T2DM participated in a CS cessation program, which generated enhancements in glycemic control and cardiometabolic risk factors over 3 months of follow-up [88]. In addition, younger patients displayed considerable improvement in HbA1c and total cholesterol levels than older patients; by contrast, patients with a low amount of baseline CS demonstrated improvements in diastolic blood pressure. Fasting plasma glucose was found to be an independent risk factor for survival in patients with advanced non-small-cell lung carcinoma after chemoradiotherapy treatment [89]. These clinical results suggest that CS cessation can sustain good health among smokers, reducing systemic inflammation in the lung–gut axis and delaying CS-associated DM progression.

### 8.2. High-Fiber Diet and Probiotics Administration

Dietary fiber consumption was significantly and inversely correlated with the risk of colorectal polyps, particularly colorectal adenomatous polyps, in smokers [90]. The results of a clinical trial indicated that probiotics alleviated dyslipidemia and improved metabolic health in patients with T2DM [91]. Moreover, a high–plant-derived fiber diet was related to the enrichment of SCFA-generating microbiota and thus increased SCFA production due to bacterial fermentation of the fibers [92]. Diets with lower fiber and higher fat and sugar levels reduced the number of the beneficial *Firmicutes* bacteria, which ferment dietary plant-derived polysaccharides to SCFAs, increasing mucosa-related *Proteobacteria* spp. [93]. Consumption of high-fiber diets containing fermentable pectin and nonfermentable cellulose prevented emphysema pathogenesis through the modulation of gut microbiota metabolism and population. Emphysema mice on a high-cellulose diet demonstrated the highest abundance of the class *Alphaproteobacteria* (specifically *Proteobacteria*) and the phylum *Verrucomicrobia* (specifically *Akkermansia*), which has numerous beneficial biological functions for host health [75]. In addition, feces from the emphysema with a high-fiber (high-cellulose and pectin) diet group demonstrated a significant abundance of acetate, butyrate, and propionate [75]. Metabolism of dietary fiber by gut microbiota with subsequent production of short-chain fatty acids, such as butyrate, propionate, and acetate, may be a key mechanism by which microbes regulate airway inflammation. High-fiber diets increase circulating SCFA levels and alleviate allergic lung inflammation, whereas low-fiber diets downregulate SCFA production associated with allergic airway disease progression [94]. The populations of bacteria from *Lachnospiraceae* and *Bacteroidaceae*, which metabolize dietary fiber into SCFAs, were enriched after consumption of the fecal microbiota transplantation (FMT) and high-fiber diet, which decreased emphysema development [95]. The high-fiber diet induced SCFA-producing microbiota and connected to elevated levels of glucagon-like peptide-1, a decreased level of acetylated hemoglobin, and enhanced blood-glucose regulation [96]. These results highlight that a high-fiber diet can modulate microbial community structures and effectively alter gut microbiota ecosystems and prevent systemic inflammation and metabolic disorders. Thus, treatment with probiotics may suppress CS-induced systemic inflammation and gut dysbiosis through the lung–gut axis.

As indicated in Table 1, growing evidence has confirmed the benefit of probiotics for CS or PM-mediated lung inflammatory diseases and metabolic disorders. In mice, probiotic Prato cheese (including *Lactobacillus* (*L*.) *casei* 01) consumption reduced CS-induced oxidative stress in the lungs, gut, and liver; it also reduced leukocyte numbers in bronchoalveolar lavage fluid as well as lipid peroxidation levels [97]. *L. rhamnosus* consumption reduced CS-mediated lung-infiltrating cell number, caused alveolar enlargement, and worsened collagen deposition and mucus secretion [98]. Administration of a probiotic including *L. acidophilus*, *Enterococcus* (*E*.) *faecium*, *Bacillus* (*B*.) *subtilis*, and *Bifidobacterium* (*B*.) *bifidum* increased mandibular bone mineral density in CS-exposed rats [99]. The SCFA-generating probiotic *L*. *plantarum* 299v led to decreased systolic blood pressure, plasma IL-6 levels, and lipid oxidation in current smokers [100]. The probiotic mixture VSL#3 (including *L. plantarum*, *L. casei*, *L. acidophilus*, *L. delbrueckii* subsp. *Bulgaricus*, *Bifidobacteria* (*B.*) *infantis*, *B. longum*, *B. breve*, *and Streptococcus* (*S*.) *salivarius* subsp. *Thermophilus*) downregulated intestinal permeability and reduced colorectal cancer tissue numbers in APC^min/+^ mice exposed to CS [101,102]. *L. paracasei* L9 inhibited asthma caused by PM exposure by rebalancing the Th1/Th2 immune response and downregulating the IL-17-based proinflammatory immune response [103]. Besides, supplemental with a combination of the probiotic-*Bifidobacterium* (*B*.) *lactis* BB-12 and nutritional intervention (including anti-oxidants-(Vitamin C/E) and the anti-inflammatory docosahexaenoic acid) could alleviate PM_2.5_-induced lung inflammation [104]. Daily *L*. *casei Shirota* (LcS) intake significantly enhanced NK cytotoxic activity and CD16^+^ cell number in male smokers when compared with placebo controls [105]. Probiotics-Ecologic^®^ Barrier 849 (including *Bifidobacterium (B*.*) bifidum* W23, *B. lactis* W51, *B. lactis* W52, *L*. *casei* W56, *L*. *salvarius* W24, *L*. *acidophilus* W37, *L. brevis* W63, *Lactococci lactis* (Lc) W19, and *Lc*. *lactis* W58) with the protective effect on the regulating inflammatory responses and intestines integrity following inhaled diesel exhaust particles [106]. *L. Rhamnosus* GG (LGG) intervention was identified to have beneficial effects on the PM_2.5_ mediated pathological damage, and the mechanism was linked to the decreasing of the inflammatory response, regulation of the balance of Th17 and Treg cells and preservation of the stabilities of the internal intestinal environment [5]. Probiotic *L. paracasei* NL41 ameliorated insulin resistance, oxidative stress, and beta-cell function in a T2DM rat model [107]. After 12 weeks of treatment with a probiotic mixture containing five SCFA-generating strains (i.e., *B. infantis*, *Clostridium butyricum*, *Clostridium beijerinckii*, *Akkermansia muciniphila*, and *Anaerobutyricum hallii*), a significant improvement in postprandial glucose control was noted in patients with T2DM [85]; moreover, after probiotic treatment, circulating butyrate and ursodeoxycholate levels increased in patients with T2DM. Thus, probiotics administration could rescue CS- or PM-mediated inflammation and metabolic disorders.

### 8.3. SCFAs Supplements and Regulation of Systemic Inflammation

As shown in Figure 2, SCFAs are crucial gut microbiota-derived carboxylic acids containing up to six carbons and thus are essential for maintaining intestinal homeostasis. SCFAs produced by microbiota mainly through dietary fiber fermentation include acetate (two carbons), propionate (three carbons), and butyrate (four carbons) [72]. These compounds enter circulation and target peripheral organs, where they regulate the immune system and cellular metabolism and reduce inflammation. Activated SCFA signaling can strengthen the gut barrier function. Recent findings have, however, demonstrated that SCFAs are vital in intestinal and cardiac inflammation modulation [72,82]. In recent years, SCFAs have been reported to control two major signaling pathways, each containing G-protein-coupled receptors (GPRCs) and histone deacetylases (HDACs) [82]. SCFAs are, therefore, vital for maintaining gut health, and the reduction in SCFA levels affects metabolism in peripheral tissues. SCFAs protect the gut barrier from disruption by reducing LPS–NLRP3 inflammasome signaling through HDAC activity inhibition [108]. GPCRs, such as GPR41 (FFAR3), GPR43 (FFAR2) OLFR78, and GPR109A, interact with SCFAs such as butyrate, acetate, and propionate with different binding activities. Propionate and acetate activate GPR43 signaling, whereas butyrate and propionate activate GPR41 axis signaling [109].

Given the high prevalence of CS-mediated inflammatory and metabolic disorders, the development of relevant therapeutics targeting the lung–gut axis is warranted. As depicted in Figure 2, CS- or PM-mediated downregulation of SCFAs and the SCFA-generating microbiota population can inhibit systemic inflammation and metabolic disorders. Exposure of mouse gut mucosa to cigarette smoke compounds has been noted to alter the microbial response and induce inflammation in the intestines [57]. Chronic PM exposure induces COPD pathological progression, lung inflammation, and gut microbiota dysbiosis; it also increases LPS and reduces SCFA levels [58]. Nevertheless, treatment with SCFAs may stimulate the GPR41/43 axis and reduce HDAC levels, thus inhibiting inflammation [110]. Bufei Jianpi formula (BJF), a type of traditional Chinese medicine, alleviates lung inflammation and improves lung function, significantly increasing the *Firmicutes* population, the *Firmicutes*-to-*Bacteroidetes* ratio, and SCFA levels [111]. BJF also downregulates the expression of the NLRP3/Caspase-1/IL-8/IL-1β axis by upregulating SCFA/GPR43 signaling [111]. In ovalbumin (OVA)-challenged mice, butyrate administration could alleviate lung inflammation and mucus generation, with reduced numbers of lung-infiltrated eosinophils and Th9 cells [112]. In mice, sodium butyrate was noted to prevent lung ischemia-reperfusion injury by inhibiting JAK2/STAT3 and NF-κB signaling, thus alleviating inflammation and reducing oxidative stress [113]. Free fatty acid receptors, such as GPR40 (FFAR1) and GPR120 (FFAR4), are abundantly expressed in lung epithelial cells. A recent study demonstrated the therapeutic efficacy of targeting FFAR4 against bronchoconstriction associated with inflammatory airway disease [114]. SCFAs can mediate the functions of adipose, skeletal muscle, and liver tissues and improve glucose homeostasis and insulin sensitivity [115]. In addition, circulating SCFA levels are positively associated with fasting concentrations of glucagon-like peptide-1 and are negatively associated with triacylglycerol and free fatty acid whole-body lipolysis (glycerol) levels [116]. SCFA-butyrate could partially improve T2D-induced kidney injury through GPR43-mediated inhibition of NF-κB signaling and oxidative stress, recommending butyrate might be a possible therapeutic agent in ameliorating DM-related comorbidity [117]. Oral acetate treatment alleviated nicotine-induced IR, glucose intolerance, and cardiorenal lipotoxicity through the inhibition of uric acid generation and suppression of creatine kinase activity in rats [118]. Thus, targeting SCFA/GPCR axis may lessen the air pollution mediated inflammation and metabolic disorder.

## 9. Conclusions

The lung–gut axis is a valuable therapeutic target for lung inflammation diseases and associated metabolic disorders. CS mediates systemic inflammation and metabolic disorders through the lung–gut axis and thus worsens COPD, metabolic disorders, and DM pathogeneses. Either CS or PM exposure can cause microbiota dysbiosis, leading to decreased gut barrier function, inflammation, glucose metabolic dysregulation, IR, and DM progression. CS cessation, along with the consumption of high dietary fiber, probiotics, and SCFAs, as a therapeutic strategy, can alleviate CS- or PM-mediated chronic inflammation and metabolic disorders. Thus, targeting CS- or PM-mediated inflammation and metabolic disease in the lung–gut axis can inhibit lung inflammation diseases and their comorbidities. Thus, targeting the lung–gut axis may reduce the occurrence and severity of CS- or PM-mediated inflammatory and metabolic disorders or both.

Thus far, however, studies have reported on (1) the effects of distinct probiotic strains in PM-induced inflammatory and metabolic disorders, (2) the prevention of PM accumulation in the human body, or (3) the effects of dietary fiber consumption or SCFAs supplements as a therapeutic strategy for PM-associated diseases. Further research on probiotic consumption for gut microbiota homeostasis and systematic inflammation after exposure to air pollution is warranted. Specific components associated with PM, such as chemical substances or certain microbial entities, also need to be clarified during disease progression. Moreover, in vivo studies and clinical trials evaluating whether interventions targeting microbiota homeostasis and thus regulating the SCFA/GPCR axis can alleviate adverse lung inflammation events and prevent its comorbid metabolic disorders are required.

## Figures and Tables

**Figure 1 cells-12-00901-f001:**
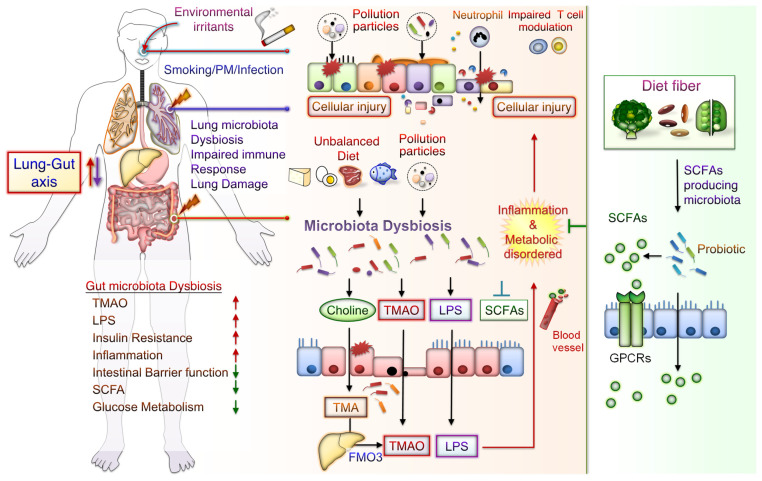
Potential Effects of CS or PM exposure on the lung–gut axis, leading to aggravating systemic inflammation and metabolic disorders. Cigarette smoking (CS) and ambient particulate matter (PM) are critical factors mediating gut microbiota dysbiosis and changes in microbiota-derived metabolites. Microbiota endotoxins and metabolites, such as lipopolysaccharide (LPS), metabolite trimethylamine (TMA), N-oxide (TMAO), and short-chain fatty acid (SCFA), participate in the regulation of system inflammation, which influences lung function. CS or PM inhalation leads to gut microbiota dysbiosis. Gut dysbiosis may, in turn, increase the levels of TMA—which is then converted to TMAO by the liver enzyme flavin-containing monooxygenase 3 (FMO3). Dietary fiber fermentation by distinct SCFA-associated gut microbiota leads to SCFA generation. Activation of SCFA/G-protein-coupled receptor (GPCR) signaling is crucial for maintaining gut barrier function and inhibiting inflammation. Thus, targeting the lung–gut axis may prevent systemic inflammation and metabolic disease.

**Figure 2 cells-12-00901-f002:**
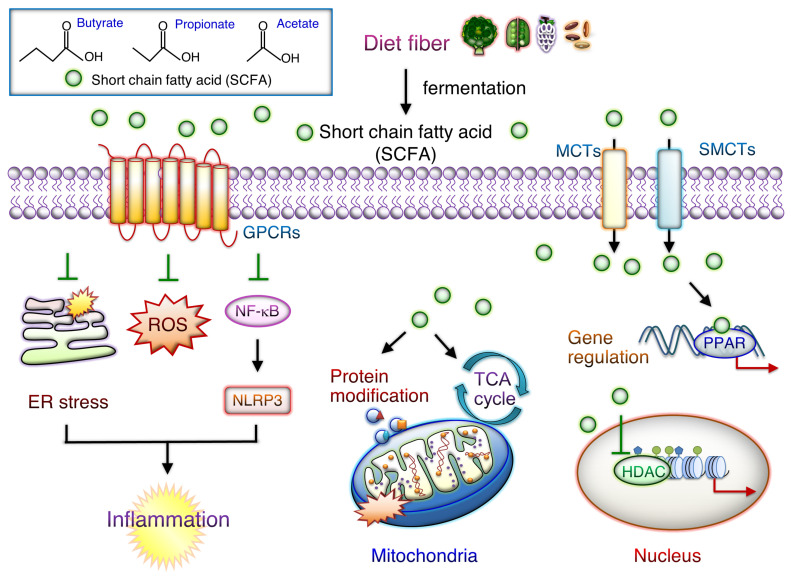
Potential function of SCFAs-mediated cellular downstream signaling. Short-chain fatty acids (SCFAs) bind to G-protein-coupled receptors (GPRCs) on cell membranes and activate downstream signaling. SCFAs also can enter the cytoplasm either through passive diffusion or membrane transporters, including monocarboxylate transporter 1 (MCT1) and sodium-coupled MCT 1 (SMCT1). GPCRs (e.g., GPR40, GPR41, GPR42, GPR43, GPR91, GPR109A, GPR120, OLFR78, and OLFR78) play roles in inhibiting inflammation and electrical remodeling as well as reducing mitochondrial reactive oxygen species and endoplasmic reticulum (ER) stress levels. In cells, SCFAs have epigenetic effects by regulating downstream cell signaling pathways and thus serve as cellular metabolism substrates. They thus inhibit histone deacetylase (HDAC) activities. SCFAs participate in energy metabolism by regulating mitochondrial proteins and participate in the tricarboxylic acid cycle.

**Table 1 cells-12-00901-t001:** Therapeutic effects of probiotics on CS- and PM-mediated inflammation and metabolic disorders.

Probiotic	Study Design	Tissue/Outcomes	Proposed Mechanisms	Ref.
*L*. *casei* 01	Male C57BL/6 mice were grouped as CS (12 cigarettes per day) + regular chow, CS + conventional cheese (daily), CS + probiotic cheese (including *L. casei* 01, 3 × 10^8^ to 3 × 10^9^ CFU/30 g/day), all for 5 days.	**Lung:** Oxidative stress ↓ Lipid peroxidation ↓ (lung and BAL) Leukocyte count (BAL) ↓ **Gut:** Oxidative stress ↓ **Liver:** Oxidative stress ↓	Probiotic cheese reduces CS-induced oxidative stress in the lungs, gut, and liver.	[97]
*L. rhamnosus*	C57Bl/6 mice were grouped as control (pure air inhalation for 60 days), COPD (14 CS inhalations for 60 days), *L. rhamnosus* (10^9^ CFU/3 days/week) + COPD.	**Lung:** Lung remodeling ↓ Pro-Inflammatory mediators ↓ Anti-inflammatory mediator ↑	*L. rhamnosus* mediates the balance between pro-inflammatory and anti-inflammatory cytokines to control airway and lung remodeling in COPD.	[98]
*L*. *acidophilus*, *E. faecium*, *Bacillus subtilis*, and *B*. *bifidum*	Rats were divided into control (without CS or probiotic), probiotic, CS, and CS + probiotic groups. The probiotic included *Lactobacillus acidophilus*, *Enterococcus faecium*, *Bacillus subtilis*, and *Bifidobacterium bifidum* at 1 × 10^9^, 2.1 × 10^9^, 2.9 × 10^9^, and 2 × 10^9^ CFU, respectively, incorporated in 1 kg of feed. It was administered for 6 months.	**Bone:** Bone mineral density ↑	Probiotic intervention increased bone mineral density and consequently exerted a protective effect on mandibular bone structures in rats exposed to CS.	[99]
*L*. *plantarum* 299v	In a controlled, double-blind clinical trial, 36 healthy smokers were administered 400 mL of a rose-hip drink with *L. plantarum* 299v (5 × 10^7^ CFU/mL) daily for 6 weeks; the control group also received the same volume of drink but without the probiotic for 6 weeks.	**Peripheral Blood:** Systolic blood pressure ↓ Plasma IL-6 ↓ Lipid peroxidation and oxidative stress ↓	In smokers, *L. plantarum* 299v treatment reduces systolic blood pressure, inflammation, and lipid oxidation, thus alleviating cardiovascular risk factors.	[100]
VSL#3 (*L. plantarum, L. casei, L. acidophilus, L. delbrueckii subsp. Bulgaricus, B. infantis, B. longum, B. breve, and S*. *salivarius subsp. Thermophilus*)	APC^min/+^ or wild-type mice were exposed to smoke from five cigarettes per day for 4, 8, or 12 weeks. Subsequently, 15 mg of VSL#3 (Grifols, Barcelona, Spain [101]) was dissolved in 200 µL of PBS and administered through oral gavage. Intestinal permeability and APC^min/+^-associated colorectal cancer cell growth were analyzed.	**Gut:** Colorectal cancer tissue numbers ↓ Intestinal permeability ↓	VSL#3 reduces CRC cell population and intestinal permeability in APC^min/+^ mice exposed to CS.	[102]
*L*. *paracasei* L9	Mice with asthma [induced through 21 days of ovalbumin (OVA) sensitization and challenge model] were exposed to PM_2.5_ (from an urban area of Beijing) twice on the last challenge, followed by orally *Lactobacillus paracasei* L9 administration at 4 × 10^9^ CFU/mouse daily.	**Lung:** Airway hyperresponsiveness (AHR) ↓ Eosinophil/neutrophil ↓ Th2-related cytokines ↓ Th1 related IFN-γ ↑ IL-17A ↓	Oral administration of *L. paracasei* L9 diminishes PM_2.5_-mediated enhancement of the airway allergic response and hyperresponsiveness in a mouse model of asthma.	[103]
*B. lactis* BB-12	Female BALB/c mice received pharyngeal aspiration with either sham treatment or PM_2.5_-containing aerosols. Before treatment, mice were fed either a regular chow or a supplemental diet (including docosahexaenoic acid, vitamins C and E, and *B. lactis* BB-12).	**Lung:** Inflammation ↓	Supplemental diet reduces PM_2.5_-induced lung inflammation.	[104]
LcS	In a placebo-controlled, double-blind clinical trial, 72 healthy male smokers were randomly divided for the intake of LcS powder (4 × 10^10^ CFU/daily) or placebo for 3 weeks.	**Peripheral blood:** NK cell activity ↑ CD16^+^ cell number ↑	LcS intake in male smokers is associated with decreased cytotoxic activity and CD16^+^ cell numbers.	[105]
Ecologic^®^ Barrier 849 (*B*. *bifidum* W23, *B*. *lactis* W51, *B*. *lactis* W52, *L*. *casei* W56, *L*. *salvarius* W24, *L*. *acidophilus* W37, *L*. *brevis* W63, *Lc*. W19, and *Lc*. *lactis* W58)	Mice with a high-fat diet were grouped as control (OA exposure of sterile saline 2 times a week for 28 days), PM (OA exposure of 35 µg diesel exhaust particles 2 times a week for 28 days), PM+ Ecologic^®^ Barrier 849 (~7.5 × 10^8^ CFU/days/28 days).	**Gut:** Intestinal integrity ↑ Inflammation ↓	Probiotics-Ecologic^®^ Barrier 849 with the protective effect on regulating inflammatory responses and intestines integrity following inhaled diesel exhaust particles.	[106]
LGG	Rats were grouped as control (normal air exposure for 112 days), PM_2.5_ (exposure in PM_2.5_ chamber for 112 days), PM_2.5_+ LGG group (10^9^ CFU/day from day 84 to day 112).	**Lung:** Inflammation ↓ Th17/Treg balance ↑ **Gut:** Beneficial bacteria ↑ Bacteria associated with inflammation ↓	LGG intervention with preventive effects on the PM_2.5_-induced inflammatory response, Th17/Treg imbalance and intestinal internal environment instability.	[5]

REF., References; *L. casei* 01, *Lactobacillus casei* 01; *L. rhamnosus*, *Lactobacillus rhamnosus*; *L. acidophilus*, *Lactobacillus acidophilus*; *E. faecium*, *Enterococcus faecium*, *B. subtilis*, *Bacillus subtilis*; *B. bifidum*, *Bifidobacterium bifidum*; *L. plantarum*, *Lactobacilli plantarum*; *L. acidophilus*, *Lactobacilli acidophilus*; *L. casei*, *Lactobacilli casei*; *L. casei delbrueckii* subsp. *Bulgaricus*, *Lactobacilli casei delbrueckii* subsp. *Bulgaricus*; *B. infantis*, *Bifidobacteria infantis*, *B. breve*, *Bifidobacteria breve*, *B. longum*, *Bifidobacteria longum*, *S. salivarius* subsp. *Thermofilus*, *Streptococcus salivarius* subsp. *Thermofilus*; *L. paracasei* L9, *Lactobacillus paracasei* L9; *B. lactis* BB-12, *Bifidobacterium lactis* BB-12; LcS, *Lactobacillus casei* Shirota; *B. bifidum* W23, *Bifidobacterium* (*B*.) *bifidum* W23; *B. lactis* W51, *Bifidobacterium lactis* W51; *B. lactis* W52, *Bifidobacterium lactis* W52; *L. acidophilus* W37, *Lactobacillus acidophilus* W37; *L. brevis* W63, *Lactobacillus brevis* W63; *L. casei* W56, *Lactobacillus casei* W56; *L. salvarius* W24, *Lactobacillus salvarius* W24; *Lc. lactis* W19, *Lactococci lactis* W19; *Lc. lactis* W58, *Lactococci lactis* W58; LGG, *Lactobacillus rhamnosus* GG; CS, cigarette smoking; BALF, bronchoalveolar lavage fluid; CFU, colony-forming units; PBS, phosphate-buffered saline; OA, oropharyngeal aspiration; ↑, increase; ↓, decrease.

## Data Availability

Not applicable.

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
