# Peer review of "Targeting Lung–Gut Axis for Regulating Pollution Particle–Mediated Inflammation and Metabolic Disorders"

_cells, 2023, doi:10.3390/cells12060901_

Round 1

Reviewer 1 Report

The review entitled "targeting gut-lung axis for regulating pollution particle-mediated inflammation and metabolic disorders" by Cheng et al addresses a relevant topic within the environmental toxicology for which the authors have performed an extensive literature search. However, the structure of the review makes it complex and sometimes not logical for the reader.

Major comments:

1. The structure of the review is complex, especially without a table of content and a short introduction of the topic, aim/RQ discussed including the setup of the review. Instead, there are 9 chapters without a clear overview how they are related or what will or will not be addressed additionally in the next chapters. This makes it difficult for the reader to understand what the authors want to demonstrate with this review. Comprehensive restructuring of the review and adding a general introduction of the topic and setup of the review is warranted.

2. Figure 2 is an extended version of Figure 1 and therefore Figure 1 can be omitted. Moreover, Figure 1 is misleading as that now indicates that PM and CS can only induce lung diseases via affecting the gut which is of course rather strange considering the direct pulmonary exposure to these inhaled toxicants.

3. In relation to the previous comment, I would recommend the authors to reconsider the setup of the review and to flip the axis into lung-gut axis. It does not make sense to start with CS/PM-induced gut problems leading to pulmonary issues whereas the other way around would be more logical. In that way, it would also make more sense that the involvement of the gut would aggrevate the pulmonary responses and damage and the possible health beneficial effects of dietary interventions would then also be nicely introduced.

4. The table contains mostly pulmonary studies (which fits with the previous 2 comments) and should be better organized and formatted to clearly make the distinction between pulmonary and gut effects/studies and how that supports the claims made by the authors.

5. It took me until the conclusion to fully understand the entire picture painted by the authors on this topic and what their goal with this review was. Adding an introduction to this review (on top of a new structure, see comment 1) that relates to this nice conclusion would increase the readability of this review.

Minor comments:

1. chapter 4 is missing, instead two chapters 3 are in

2. 9.3 reads suppliments whereas it should be supplements

Author Response

Dear Reviewer,

Thank you for your detailed comments, which were highly helpful in revising this manuscript.

Comments and Suggestions for Authors

Response to Reviewer#1

The review entitled "targeting gut-lung axis for regulating pollution particle-mediated inflammation and metabolic disorders" by Cheng et al addresses a relevant topic within the environmental toxicology for which the authors have performed an extensive literature search. However, the structure of the review makes it complex and sometimes not logical for the reader.

Major comments:

  1. Regarding to the comments “The structure of the review is complex, especially without a table of content and a short introduction of the topic, aim/RQ discussed including the setup of the review. Instead, there are 9 chapters without a clear overview how they are related or what will or will not be addressed additionally in the next chapters. This makes it difficult for the reader to understand what the authors want to demonstrate with this review. Comprehensive restructuring of the review and adding a general introduction of the topic and setup of the review is warranted.”

    Answer: Thanks for your suggestions. We have added a short introduction of the topic and setup of the review in the revised manuscript (page 1, lines 39-70) as below. Additionally, we added more contextually linked descriptions at the end or beginning of the section to make it more readable.

 “1. The Lung-Gut Axis in Lung and Metabolic Disease

The homeostasis of the gut microbiota and its’ metabolites act as a critical role in the alternation of locally and systemically immune system [1]. Accumulating studies point to an important interaction between the gut microbiota and the lung, termed the "lung–gut axis" [1]. Furthermore, increasing evidence declares its crucial role in maintaining host metabolic homeostasis and in the pathogenesis of lung diseases [2]. Particulate air pollution components such as hazardous substance and metals that remain in the lungs can lead to dysbiosis of the lung and gut microbiota, resulting in decreased lung function [3]. The role of the lung–gut axis is related to the action of microbiota and gut microbe-derived components and metabolites, such as lipopolysaccharide (LPS), metabolite trimethylamine (TMA) N-oxide (TMAO), and short-chain fatty acid (SCFA), as main regulators for the immune and metabolic systems [4-6]. SCFAs regulate immune homeostasis and systemic immune responses, such as modulation of pro-inflammatory or anti-inflammatory [7]. Notably, gut microbiota dysbiosis is one of the risk factors promoting insulin resistance (IR) in type 2 diabetes mellitus (T2DM) [8]. Current evidence suggests that the chemical constituents of the air pollutant mixture may affect the pathogenesis of T2DM [9]. However, there is a need to elucidate the detailed mechanisms by which air pollutants cause metabolic dysfunction, IR, and T2DM, including increased inflammation, oxidative stress, and endoplasmic reticulum stress [10]. Of note, emerging experimental and epidemiological findings highlight a key factor that exposure to pollutant particles induces alterations in the gut microbiota and systemic metabolism, which may contribute to glucose metabolism disorder, IR, and T2DM. [11]. In this while, the regulation of gut microbiota homeostasis or probiotics administration may act as a vital role on the regulation of lung-gut axis. Indoor and outdoor air pollution such as cigarette smoking (CS) and ambient particulate matter (PM) increase lung dysfunction and risk of chronic obstructive pulmonary disease (COPD) and respiratory death [12]. This review includes recent studies on the effects of CS and ambient PM on gut microbiota dysbiosis, systemic inflammation, and metabolic diseases possibly due to disruption of the lung-gut axis. Furthermore, we discuss the potential prevention of particulate pollution-mediated dysregulation of the lung-gut axis and consequent microbial-derived metabolites may reduce the risk of metabolic dysfunction and associated disease progression.”

  1. Regarding to the comments “Figure 2 is an extended version of Figure 1 and therefore Figure 1 can be omitted. Moreover, Figure 1 is misleading as that now indicates that PM and CS can only induce lung diseases via affecting the gut which is of course rather strange considering the direct pulmonary exposure to these inhaled toxicants.”

Answer: Thanks for your suggestion, and Figure 1 has been removed to avoid misunderstanding of the content.

  1. Regarding to the comments “In relation to the previous comment, I would recommend the authors to reconsider the setup of the review and to flip the axis into lung-gut axis. It does not make sense to start with CS/PM-induced gut problems leading to pulmonary issues whereas the other way around would be more logical. In that way, it would also make more sense that the involvement of the gut would aggrevate the pulmonary responses and damage and the possible health beneficial effects of dietary interventions would then also be nicely introduced.”

Answer: We greatly appreciate this comment. We have changed the term of “gut-lung axis” to “lung-gut axis” in the content and figure. Air pollution-mediated dysregulation of the lung-gut axis exacerbates lung injury, metabolic dysfunction and possible beneficial health effects of dietary interventions have been included in the section of introduction (page 1, lines 39-70) as follows.

1. The Lung-Gut Axis in Lung and Metabolic Disease

The homeostasis of the gut microbiota and its’ metabolites act as a critical role in the alternation of locally and systemically immune system [1]. Accumulating studies point to an important interaction between the gut microbiota and the lung, termed the "lung–gut axis" [1]. Furthermore, increasing evidence declares its crucial role in maintaining host metabolic homeostasis and in the pathogenesis of lung diseases [2]. Particulate air pollution components such as hazardous substance and metals that remain in the lungs can lead to dysbiosis of the lung and gut microbiota, resulting in decreased lung function [3]. The role of the lung–gut axis is related to the action of microbiota and gut microbe-derived components and metabolites, such as lipopolysaccharide (LPS), metabolite trimethylamine (TMA) N-oxide (TMAO), and short-chain fatty acid (SCFA), as main regulators for the immune and metabolic systems [4-6]. SCFAs regulate immune homeostasis and systemic immune responses, such as modulation of pro-inflammatory or anti-inflammatory [7]. Notably, gut microbiota dysbiosis is one of the risk factors promoting insulin resistance (IR) in type 2 diabetes mellitus (T2DM) [8]. Current evidence suggests that the chemical constituents of the air pollutant mixture may affect the pathogenesis of T2DM [9]. However, there is a need to elucidate the detailed mechanisms by which air pollutants cause metabolic dysfunction, IR, and T2DM, including increased inflammation, oxidative stress, and endoplasmic reticulum stress [10]. Of note, emerging experimental and epidemiological findings highlight a key factor that exposure to pollutant particles induces alterations in the gut microbiota and systemic metabolism, which may contribute to glucose metabolism disorder, IR, and T2DM. [11]. In this while, the regulation of gut microbiota homeostasis or probiotics administration may act as a vital role on the regulation of lung-gut axis. Indoor and outdoor air pollution such as cigarette smoking (CS) and ambient particulate matter (PM) increase lung dysfunction and risk of chronic obstructive pulmonary disease (COPD) and respiratory death [12]. This review includes recent studies on the effects of CS and ambient PM on gut microbiota dysbiosis, systemic inflammation, and metabolic diseases possibly due to disruption of the lung-gut axis. Furthermore, we discuss the potential prevention of particulate pollution-mediated dysregulation of the lung-gut axis and consequent microbial-derived metabolites may reduce the risk of metabolic dysfunction and associated disease progression.”

  1. Regarding to the comments “The table contains mostly pulmonary studies (which fits with the previous 2 comments) and should be better organized and formatted to clearly make the distinction between pulmonary and gut effects/studies and how that supports the claims made by the authors.”

Answer: We greatly appreciate this comment. In the revised Table 1, we have made more clear distinction between lung and/or gut effects/studies.

  1. Regarding to the comments “It took me until the conclusion to fully understand the entire picture painted by the authors on this topic and what their goal with this review was. Adding an introduction to this review (on top of a new structure, see comment 1) that relates to this nice conclusion would increase the readability of this review.”

Answer: We greatly appreciate this comment. We have added a short introduction (page 1, lines 39-70) in this review that is relevant to topic, goal and conclusion, which increase the readability of this review.

Minor comments:

  1. chapter 4 is missing, instead two chapters 3 are in

Answer: Thanks, this error has been corrected.

  1. 9.3 reads suppliments whereas it should be supplements

Answer: Thanks, this error has been corrected.

Sincerely yours,

Sheng-Ming Wu

Division of Pulmonary Medicine,

Department of Internal Medicine,

School of Medicine, College of Medicine,

Taipei Medical University

Reviewer 2 Report

This review focusses on the gut-lung axis for regulation of cigarette smoke (CS) and ambient particulate matter (PM)-mediated inflammation and metabolic disorders. Here the authors state that CS and PM exposure are risk factors of metabolic disorders which can also cause gut microbiota dysbiosis. Interruption of the gut-lung axis is one of the potential mechanisms leading to gut microbiota dysbiosis.

Specific Comments

Comments to the Editor:

Thank you for giving me the opportunity to review this manuscript. Although the topic is interesting, the model used is questionable. A lot of the images and WB data are of poor quality which makes interpretation very difficult or almost impossible. My recommendation is to reject the manuscript.

Specific Comments:

1.       The review would benefit from a brief introductory paragraph and clarification of some terminology, e.g. the terms “metabolic disorders” and “metabolic syndrome” need to be clarified and a distinction made between those terms. This is true throughout the article, e.g. “metabolic derangement”.

2.       The review may benefit from 2 paragraphs on the effects of CS and PM on the lung and then the effects of CS and PM on the gut before mixing everything together.

3.       The review may benefit from a paragraph on inflammation, what distinguishes CS- and PM-induced lung inflammation from CS- and PM-induced lung inflammation, and then the overlapping effects where the different aspects come in.

4.       It remains unclear what comes first, the insult to the lung or the insult to the gut, or both. This is also true for the inflammation which is the link between the two. Figure 1 seems to be overly simplified. Maybe Figure 1 could be revised to highlight or emphasize the driving forces leading to gut microbiota dysbiosis.

5.       Figure 1 also lists disease states with disease markers in the same box which causes confusion.

6.       It is unclear why the authors discuss the role SCFA/GPCR signaling for regulation of systemic inflammation before going into CS cessation and its effects.

Author Response

Dear Reviewer,

Thank you for your detailed comments, which were highly helpful in revising this manuscript.

Comments and Suggestions for Authors

Response to Reviewer#2

This review focusses on the gut-lung axis for regulation of cigarette smoke (CS) and ambient particulate matter (PM)-mediated inflammation and metabolic disorders. Here the authors state that CS and PM exposure are risk factors of metabolic disorders which can also cause gut microbiota dysbiosis. Interruption of the gut-lung axis is one of the potential mechanisms leading to gut microbiota dysbiosis.

Specific Comments

Comments to the Editor:

Thank you for giving me the opportunity to review this manuscript. Although the topic is interesting, the model used is questionable. A lot of the images and WB data are of poor quality which makes interpretation very difficult or almost impossible. My recommendation is to reject the manuscript.

Answer: Thanks for your comment. We provided and uploaded high resolution images of Figures 1-2. Furthermore, this review article does not include WB data.

Specific Comments:

  1. Regarding to the comments “The review would benefit from a brief introductory paragraph and clarification of some terminology, e.g. the terms “metabolic disorders” and “metabolic syndrome” need to be clarified and a distinction made between those terms. This is true throughout the article, e.g. “metabolic derangement”.”

Answer: We greatly appreciate this comment. We have clarified the difference between the terms of "metabolic disorder" and "metabolic syndrome" in the revision (Page 2, line 80-85, red font) as listed below, and changed “Metabolic derangement” to “Metabolic disorders” (Page 5, line 194).

“In addition, CS is also considered a critical risk factor for metabolic disorder and metabolic syndrome (MetS). Metabolic disorders occur when aberrant chemical reactions in the body alter normal metabolic processes. However, MetS encompasses a spectrum of metabolic abnormalities including hypertension, obesity, IR, and atherogenic dyslipidemia, and is strongly associated with increased risk of diabetic cardiovascular disease [17].”

  1. Regarding to the comments “The review may benefit from 2 paragraphs on the effects of CS and PM on the lung and then the effects of CS and PM on the gut before mixing everything together.”

Answer: We greatly appreciate this comment. We have adjusted the content regarding the effects of CS on the lung effect (section 2, line 72-126) and CS on the gut effect (section 3, line 128-153), and then disuse the effects of PM on the lung effect in section 4, paragraph 1 (line 157-164) and PM on the gut effect in section 4 paragraph 2 (line 165-193) as follow sections.

2. CS Exposure induces Lung Inflammation and Diabetes Mellitus Progression

CS is a complex mixture including various compounds with suspended PM and gases [13]. Indoor PM2.5 (PM with aerodynamic diameter ≤ 2.5 micrometers) was the most reliable indicator in CS [14]. CS exposure can cause chronic inflammatory lung disease, lung infection, and predispose individuals to acute lung injury [15]. Airway epithelial cells and alveolar macrophages exposed to CS alter inflammatory signaling leading to the recruitment of eosinophils, neutrophils, lymphocytes, and mast cells to the lung—depending on the activation of different signaling pathways, such as Nuclear factor-kB (NF-kB), c-Jun N-terminal kinase, p38, and STAT3 [16]. In addition, CS is also considered a critical risk factor for metabolic disorder and metabolic syndrome (MetS). Metabolic disorders occur when aberrant chemical reactions in the body alter normal metabolic processes. However, MetS encompasses a spectrum of metabolic abnormalities including hypertension, obesity, IR, and atherogenic dyslipidemia, and is strongly associated with increased risk of diabetic cardiovascular disease [17]. Notably, CS may induce MetS, which is characterized by reduced insulin sensitivity, increased IR and plasma triglyceride level, and mediated hyperglycemia [18]. Several clinical studies have indicated a close link between MetS and lung disease [19]. MetS comprises a cluster of risk factors specific for lung inflammation diseases, such as obesity, high blood pressure, diabetes mellitus (DM), and chronic kidney disease (CKD) [20, 21]. CS exposure is strongly associated with an increased risk of progression of various diseases, including DM, cardiovascular disease, and COPD. However, prolonged CS cessation can reduce these risks.

A meta-analysis of prospective observational studies study revealed that there is a linear dose-response relationship between CS and the risk of T2DM; however, the risk is steadily declined after smoking cessation [22]. There was a dose-response relationship between HbA1c (blood sugar/glucose test) levels and CS exposure of current smokers, and there was an inverse association of the number of years since smoking cessation with HbA1c levels has been detected among ex-smokers [23]. The adverse effects of CS and hyperglycemia aggravate vascular damage in patients with DM [24]. Eliasson et al. reported a dose-dependent correlation between the per-day CS quantity and IR degree [25]. Acute CS can impair glucose tolerance and insulin sensitivity as well as increase serum triglyceride and cholesterol levels, blood pressure, and heart rate [26], whereas CS cessation can improve insulin sensitivity [27]. Moreover, CS is an independent predictor of T2DM, and CS cessation can reduce the risk of MetS [28]. Serious pulmonary events (SPEs) reduced exercise capacity and lung dysfunction and might be a clinical indicator of pre-DM or undiagnosed DM [24]. CS cessation can reduce SPE risk, morbidity, and mortality in people with DM. For instance, in DM mice, CS exposure was reported to accelerate edema and inflammatory progression, whereas CS cessation alleviated CS-mediated metabolic disorder and IR [27]. Also, DM is strongly correlated with pulmonary complications including COPD, fibrosis, pneumonia, and lung cancer [29]. In patients with COPD combined with DM, metformin—the first-line drug for T2DM—can increase inspiratory muscle strength, leading to dyspnea and COPD alleviation as well as health status and lung function improvements [30, 31]. Although the strong association of DM with inflammation-related chronic lung disease, particularly asthma and COPD, has been demonstrated epidemiologically and clinically, the underlying mechanism and pathophysiology remain unclear [32]. Numerous mechanisms have been reported; most of them have implicated the association of lung disease with DM-related inflammatory properties or pulmonary microvascular and macrovascular complications [33]. DM combined with progression to pulmonary disease is characterized as a chronic and progressive disease with high mortality and extremely few therapeutic options, possibly including metformin and thiazolidinediones [32, 34, 35]. Collectively, the complex consequences mediated by CS exposure lead not only to lung inflammation but also to the progression of MetS. In the next section, we continue to review emerging studies on CS-mediated systemic inflammation and metabolic diseases associated with the gut microbiota dysbiosis.

  1. Association of CS Exposure with Gut-Derived Microbiota and Inflammatory Bowel Disease

CS-related microbiota dysbiosis may be associated with numerous inflammatory                     lung diseases (including COPD, asthma, cystic fibrosis, and allergy) and metabolic disorders (such as IR, glucose metabolic disorder, hypertriglyceridemia, and DM) [14, 36, 37]. However, the relationship of CS with gut-derived microbiota biofunction and their dietary nutrient–derived metabolites warrants elucidation. Even with improved understanding of the mechanisms underlying the progression of inflammation-associated lung disease, CS-associated metabolic disorder remains the leading cause of morbidity and mortality globally. Several epidemiological studies have indicated a high correlation between intestinal microbiota and the lungs—termed as the lung–gut axis [1]. Gut microbiota dysbiosis is associated with inflammatory bowel disease (IBD), and it influences the gut epithelial barrier function and leads to an increased immune response and chronic inflammation disease and metabolic disorder pathogenesis. Moreover, IBD is strongly linked with COPD, DM, and gut microbiota dysbiosis [38, 39]. Alterations in gut microbiota due to an imbalanced diet may lead to enhanced local and systemic immune responses. Gut microbiota dysbiosis has been linked to not only the loss of gastrointestinal tract function but also that of airway function, such as that in asthma and COPD [1]. Moreover, CS-elicited dysbiosis has a protumorigenic role in colorectal cancer. CS-associated gut microbiota dysbiosis alters gut metabolites and diminishes gut barrier function, activating oncogenic MAPK/ERK signaling in the colonic epithelium [40]. Taken together, these results indicate that CS mediates not only the dysbiosis of gut microbiota and the dysregulation of their metabolites but also systemic inflammation and metabolic dysregulation in the lung–gut axis. In addition, ambient PM-mediated systemic inflammation and metabolic disease may also be associated with dysregulation of the lung-gut axis, as reviewed later.

  1. PM Exposure triggers Lung Inflammation, Gut Microbiota Dysbiosis, and DM Progression

Chronic exposure to particulate pollution such as PM2.5 can also lead to decreased lung function, emphysematous lesions, and airway inflammation [41, 42]; and it accelerates the CS-induced alterations in COPD progression [36]. Ambient air pollution is associated with decreased lung function and increased COPD prevalence in a large cohort study [12]. Furthermore, urban PM exposure markedly increased airway inflammatory responses through activation of reactive oxygen species (ROS)-MAPK-NF-κB signaling [43]. Thus, PM exposure contributes to respiratory disease by triggering lung inflammation and increasing oxidative stress.

     Notably, recent studies have linked PM exposure to intestinal disorders such as appendicitis, irritable bowel syndrome, and IBD [44, 45]. In addition, the secretion of proinflammatory cytokines and intestinal permeability are increased in the small intestine of mice exposed to PM10. [46]. PM-mediated dysbiosis of the microbiota is also correlated with PM-mediated gut and systemic effects. Long-term exposure to PM, such as O3, NO2, SO2, PM10, and PM2.5, as well as traffic-related air pollution, has been shown to alter microbiota diversity [47]. Moreover, both PM2.5 and PM1 exposure are positively associated with the risks of fasting glucose impairment or T2DM and negatively associated with alpha diversity indices of the gut bacteria [48]. Through systematic database analyzation, air pollution exposure is a leading cause of insulin resistance and T2DM. Besides, the association between air pollution and diabetes is stronger for particulate matters, nitrogen dioxide, and traffic associated pollutants [10]. In a meta-analysis study, exposure to PM2.5 but not PM10 or NO2 is correlated with increased disease incidence in T2DM patients [49]. Particularly, chronic ambient PM2.5 exposure associated with increased T2DM risk in several Asian populations exposed to high levels of air pollution [50]. Through global untargeted metabolomic analysis, several significant blood metabolites and metabolic pathways were identified associated with chronic exposure to PM2.5, NO2, and temperature; these included glycerophospholipid, glutathione, and sphingolipid propanoate as well as purine metabolism and unsaturated fatty acid biosynthesis [51]. In mice, PM2.5 exposure was reported to lead to increased oxidative stress, glucose intolerance, IR, and gut dysbiosis and impaired hepatic glycogenesis [52]. In rats with T2DM, short-term PM2.5 exposure significantly increased IR as well as the lung levels of inflammatory factors, such as interleukin (IL) 6, monocyte chemoattractant protein 1, and tumor necrosis factor (TNF) α [39]. High levels and prolonged periods exposure to concentrated ambient PM2.5-mediated gut dysbiosis was associated with the metabolic disorder and intestinal inflammation [53]. Taken together, these findings indicate that air pollution particles not only mediate the pathogenesis of lung inflammation disease, but also increase gut microbiota dyshomeostasis and metabolic disorders, such as IR and DM.”

  1. Regarding to the comments “The review may benefit from a paragraph on inflammation, what distinguishes CS- and PM-induced lung inflammation from CS- and PM-induced lung inflammation, and then the overlapping effects where the different aspects come in.”

Answer: We greatly appreciate this comment. In the revised manuscript, we described the effects of CS on lung inflammation in sections 2 (line 73-80) and PM on lung inflammation in section 4 (line 157-164). In fact, the harmful components carried by ambient PM may vary according to the level and duration of environmental pollution exposure, leading to different degrees of lung deposition, inflammation and injury. However, either CS or PM may worsen IBD, gut microbiota dysbiosis, and microbiota-derived metabolite alterations, thus mediating systemic inflammation and metabolic disorder development by increasing LPS and TMAO levels and reducing SCFA levels. Therefore, we discuss the CS- and PM-effects in the following sections.

2. CS Exposure induces Lung Inflammation and Diabetes Mellitus Progression

CS is a complex mixture including various compounds with suspended PM and gases [13]. Indoor PM2.5 (PM with aerodynamic diameter ≤ 2.5 micrometers) was the most reliable indicator in CS [14]. CS exposure can cause chronic inflammatory lung disease, lung infection, and predispose individuals to acute lung injury [15]. Airway epithelial cells and alveolar macrophages exposed to CS alter inflammatory signaling leading to the recruitment of eosinophils, neutrophils, lymphocytes, and mast cells to the lung—depending on the activation of different signaling pathways, such as Nuclear factor-kB (NF-kB), c-Jun N-terminal kinase, p38, and signal transducer and activator of transcription 3 [16].”

“4. PM Exposure triggers Lung Inflammation, Gut Microbiota Dysbiosis, and DM Progression

Chronic exposure to particulate pollution such as PM2.5 can also lead to decreased lung function, emphysematous lesions, and airway inflammation [41, 42]; and it accelerates the CS-induced alterations in COPD progression [36]. Ambient air pollution is associated with decreased lung function and increased COPD prevalence in a large cohort study [12]. Furthermore, urban PM exposure markedly increased airway inflammatory responses through activation of reactive oxygen species (ROS)-MAPK-NF-κB signaling [43]. Thus, PM exposure contributes to respiratory disease by triggering lung inflammation and increasing oxidative stress.”

  1. It remains unclear what comes first, the insult to the lung or the insult to the gut, or both. This is also true for the inflammation which is the link between the two. Figure 1 seems to be overly simplified. Maybe Figure 1 could be revised to highlight or emphasize the driving forces leading to gut microbiota dysbiosis.

Answer: We greatly appreciate this comment. Following the suggestion of reviewer#1 as follows “Figure 2 is an extended version of Figure 1 and therefore Figure 1 can be omitted. Moreover, Figure 1 is misleading as that now indicates that PM and CS can only induce lung diseases via affecting the gut which is of course rather strange considering the direct pulmonary exposure to these inhaled toxicants.”, and we have omitted Figure 1.

  1. Figure 1 also lists disease states with disease markers in the same box which causes confusion.

Answer: We greatly appreciate this comment. We have removed Figure 1 to avoid confusing the contribution of driving forces to disease.

  1. It is unclear why the authors discuss the role SCFA/GPCR signaling for regulation of systemic inflammation before going into CS cessation and its effects.

Answer: We greatly appreciate this comment. Accumulating findings indicate that CS or PM-mediated lung injury is closely associated with alterations in gut microbiota and low production of SCFAs. Notably, SCFA/GPCR signaling may retain gut barrier function and suppress inflammation. As shown in the revised Figure 1, SCFA could attenuate microbiota dysbiosis induced lung inflammation and cellular injury through the lung-gut axis, which may prevent systemic inflammation and metabolic dysregulation. Thus, we discuss the role of SCFA/GPCR signaling before going into the section of CS cessation in the revised manuscript (page 6, line 273-294) as follows:

“Notably, disruption of the microbiota resulted in lower SCFA levels after exposure to air pollution; this effect is linked with metabolic disorders. When rats were exposed to CS for 4 weeks, the cecal levels of SCFAs, such as acetic acid, propionic acid, butyric acid, and valeric acid, decreased significantly. Moreover, Bifidobacterium spp. significantly decreased, whereas pH in caecal contents significantly increased [76]. Intragastric administration of cigarette smoke condensate (CSC) in mice caused inflammation in the intestinal mucosa, which induced alterations in Paneth cell granules and reduced their bactericidal capacity [57]. CSC exposure caused imbalance in the gut bacterial population, promoting bacterial infection and causing ileal damage, in mice. Moreover, CSC significantly increased the abundance of bacteria from Erysipelotrichaceae but considerably reduced that of Rikenellaceae. A significant decrease was also noted in the abundance of Eisenbergiella spp. (from the family Lachnospiraceae), known for its butyrate generation capacity [77]. CS cessation–related alterations in the gut microbial ecosystem has, however, been linked to weight gain in mice [78]. The concentrations of SCFAs (i.e., acetate, butyrate, and propionate) were the lowest in mice with CS-induced emphysema; on the contrary, local SCFA levels were significantly higher in the emphysema with high-fiber (pectin and cellulose) diet group than in the untreated emphysema group [75]. In diesel exhaust particles-treated mice, dysbiosis of the gut microbiota population was associated with a dose-dependent decreasing of SCFAs (butyrate and propionate) in cecal content [79]. Collectively, accumulating findings suggest that CS- or PM-mediated lung injury is closely associated with altered gut microbiota and low SCFA production.”.

Moreover, we have restructured the role of SCFAs in disease described in the content (original section 9.3). Some aspects of SCFA regulation of systemic inflammation have been moved to Section 6 to illustrate underlying mechanisms. Additional benefits or therapeutic effects of SCFA treatment are described in revised Section 8.3 (line 468-520) as follows:

8.3. SCFAs supplements and regulation of Systemic Inflammation

As shown in Figure 2, SCFAs are crucial gut microbiota–derived carboxylic acids containing up to six carbons and thus are essential for maintaining intestinal homeostasis. SCFAs produced by microbiota mainly through dietary fiber fermentation include acetate (two carbons), propionate (three carbons), and butyrate (four carbons) [72]. These compounds enter circulation and target peripheral organs, where they regulate the immune system and cellular metabolism and reduce inflammation. Activated SCFA signaling can strengthen the gut barrier function. Recent findings have, however, demonstrated that SCFAs are vital in intestinal and cardiac inflammation modulation [72, 82]. In recent years, SCFAs have been reported to control two major signaling pathways, each containing G-protein-coupled receptors (GPRCs) and histone deacetylases (HDACs) [82]. SCFAs are therefore vital for maintaining gut health, and the reduction in SCFA levels affects metabolism in peripheral tissues. SCFAs protect the gut barrier from disruption by reducing LPS–NLRP3 inflammasome signaling through HDAC activity inhibition [108]. GPCRs, such as GPR41 (FFAR3), GPR43 (FFAR2) OLFR78, and GPR109A, interact with SCFAs such as butyrate, acetate, and propionate with different binding activities. Propionate and acetate activate GPR43 signaling, whereas butyrate and propionate activate GPR41 axis signaling [109].

Given the high prevalence of CS-mediated inflammatory and metabolic disorders, the development of relevant therapeutics targeting the lung–gut axis is warranted. As depicted in Figure 2, CS- or PM-mediated downregulation of SCFAs and the SCFA-generating microbiota population can inhibit systemic inflammation and metabolic disorders. Exposure of mouse gut mucosa to cigarette smoke compounds has been noted to alter the microbial response and induce inflammation in the intestines [57]. Chronic PM exposure induces COPD pathological progression, lung inflammation, and gut microbiota dysbiosis; it also increases LPS and reduces SCFA levels [58]. Nevertheless, treatment with SCFAs may stimulate the GPR41/43 axis and reduce HDAC levels, thus inhibiting inflammation [110]. Bufei Jianpi formula (BJF), a type of traditional Chinese medicine, alleviates lung inflammation and improves lung function, significantly increasing the Firmicutes population, the Firmicutes-to-Bacteroidetes ratio, and SCFA levels [111]. BJF also downregulates the expression of the NLRP3/Caspase-1/IL-8/IL-1β axis by upregulating SCFA/GPR43 signaling [111]. In ovalbumin (OVA)-challenged mice, butyrate administration could alleviate lung inflammation and mucus generation, with reduced numbers of lung-infiltrated eosinophils and Th9 cells [112]. In mice, NaB was noted to prevent lung ischemia–reperfusion injury by inhibiting JAK2/STAT3 and NF-κB signaling, thus alleviating inflammation and reducing oxidative stress [113]. Free fatty acid receptors, such as GPR40 (FFAR1) and GPR120 (FFAR4), are abundantly expressed in lung epithelial cells. A recent study demonstrated the therapeutic efficacy of targeting FFAR4 against bronchoconstriction associated with inflammatory airway disease [114]. SCFAs can mediate the functions of adipose, skeletal muscle, and liver tissues and improve glucose homeostasis and insulin sensitivity [115]. In addition, circulating SCFA levels are positively associated with fasting concentrations of glucagon-like peptide-1 and are negatively associated with triacylglycerol and free fatty acid whole-body lipolysis (glycerol) levels [116]. SCFA-butyrate, could partially improve T2D-induced kidney injury through GPR43-mediated inhibition of NF-κB signaling and oxidative stress, recommending butyrate might be possible therapeutic agents in the ameliorate the DM-related comorbidity [117]. Oral acetate treatment alleviated nicotine-induced IR, glucose intolerance, and cardiorenal lipotoxicity through the inhibition of uric acid generation and suppression of creatine kinase activity in rats [118]. Thus, targeting SCFA/GPCR axis may lessen the air pollution mediated inflammation and metabolic disorder.”.

Sincerely yours,

Sheng-Ming Wu

Division of Pulmonary Medicine,

Department of Internal Medicine,

School of Medicine, College of Medicine,

Taipei Medical University

Round 2

Reviewer 1 Report

My comments and suggestions have been extensively incorporated in the review which in my opinion now has the desired structure and logical setup to deliver its important message. I'm happy to accept the article in its current form.

Reviewer 2 Report

No further comments. Thank you